# AVOIDING PITFALLS FOR PRIVACY ACCOUNTING OF SUBSAMPLED MECHANISMS UNDER COMPOSITION

## ABSTRACT

We consider the problem of computing tight privacy guarantees for the composition of subsampled differentially private mechanisms. Recent algorithms can numerically compute the privacy parameters to arbitrary precision but must be carefully applied.

Our main contribution is to address two common points of confusion. First, some privacy accountants assume that the privacy guarantees for the composition of a subsampled mechanism are determined by self-composing the worst-case datasets for the uncomposed mechanism. We show that this is not true in general. Second, Poisson subsampling is sometimes assumed to have similar privacy guarantees compared to sampling without replacement. We show that the privacy guarantees may in fact differ significantly between the two sampling schemes. This occurs for some parameters that could realistically be chosen for DP-SGD.

## 1 INTRODUCTION

One of the fundamental properties of differential privacy is that composing multiple differentially private mechanisms to construct a new mechanism still satisfies differential privacy. This property allows us to design complicated mechanisms with strong formal privacy guarantees such as differentially private stochastic gradient descent (DP-SGD, Song et al. (2013); Bassily et al. (2014); Abadi et al. (2016)).

The privacy guarantees of a mechanism inevitably deteriorate with the number of compositions. Accurately quantifying the privacy parameters under composition is highly non-trivial and is an important area within the field of differential privacy. A common approach is to find the privacy parameters for each part of a mechanism and apply a composition theorem (Dwork et al., 2010; Kairouz et al., 2015) to find the privacy parameters of the full mechanism. In recent years, several alternatives to the traditional definition of differential privacy with cleaner results for composition have gained popularity (see, e.g., Dwork and Rothblum (2016); Bun and Steinke (2016); Mironov (2017); Dong et al. (2019)).

Another important concept is privacy amplification by subsampling (see, e.g., Balle et al. (2018); Steinke (2022)). The general idea is to improve privacy guarantees by only using a randomly sampled subset of the full dataset as input to a mechanism. In this work we consider the problem of computing tight privacy parameters for subsampled mechanisms under composition.

One of the primary motivations for studying privacy accounting of subsampled mechanisms is DP-SGD. DP-SGD achieves privacy by clipping gradients and adding appropriate Gaussian noise to each batch. As such, we can find the privacy parameters by analyzing the subsampled Gaussian mechanism under composition. One of the key contributions of Abadi et al. (2016) was the moments accountant which gives tighter bounds for the mechanism than the generic composition theorems. Later work improved the accountant by giving improved bounds on the Renyi Differential Privacy guarantees of the subsampled Gaussian mechanism under both Poisson subsampling and sampling without replacement (Mironov et al., 2019; Wang et al., 2020).

Even small constant factors in an $(\varepsilon, \delta)$-DP budget are important. First, from the definition, such constant factors manifest exponentially in the privacy guarantee. Furthermore, when training a model privately with DP-SGD, it has been observed that they can lead to significant differences in the downstream utility, see, e.g., Figure 1 of De et al. (2022). Consequently, "saving" such a factor

in the value of $\varepsilon$ through tighter analysis can be very valuable. This has motivated a recent line of work on numerically estimating the privacy parameters (Sommer et al., 2019; Koskela et al., 2020; 2021; Gopi et al., 2021; Zhu et al., 2022).

Most privacy accounting techniques for DP-SGD assume a version of the algorithm that employs amplification by *Poisson* subsampling. That is, the batch for each iteration is formed by including each point independently with sampling probability $\gamma$. Other privacy accountants consider a variant where random batches of a fixed size are selected for each step. Note that both of these are inconsistent with the standard method in the non-private setting, where batches are formed by randomly permuting and then partitioning the dataset. Indeed, the latter approach is much more efficient, and highly-optimized in most libraries. Consequently, many works in private ML implement a method with the conventional shuffle-and-partition method of batch formation, but employ privacy accountants that assume some other method of sampling batches. The hope is that small modifications of this sort would have negligible impact on the privacy analysis, thus justifying privacy accountants for a setting which is *technically* not matching.

The central aim of our paper is to clarify some common problems with privacy accounting in order to facilitate more faithful comparisons between DP-SGD algorithms. We organize our paper as follows.

- In Sections 4 and 5, we establish that the datasets used to compute the privacy guarantees of typical subsampled mechanism do not in general give the correct result when the mechanism is composed multiple times. Some popular privacy accountants assume otherwise, which is an error that can be corrected easily.

- In Section 6, we show that rigorous privacy accounting is *significantly* affected by the method of sampling batches. This results in sizeable differences in the resulting privacy guarantees for settings which were previously treated as interchangeable by prior works. Consequently, we demonstrate the invalidity of the common practice of using one method of batch sampling and employing the privacy accountant for another.

- Lastly, in Section 7, we discuss issues that arise in tight privacy accounting under the substitution relation. It is known for the add/remove relation that the privacy guarantees are determined by just one of two pairs of datasets. We show that this is unfortunately not the case in general under the substitution relation.

## 2    PRELIMINARIES

Differential privacy is a rigorous privacy framework introduced by Dwork et al. (2006). Differential privacy is a restriction on how much the output distribution of a mechanism can change between any pair of datasets that differ only in a single individual. Such datasets are called neighboring, and we denote a pair of neighboring datasets as $D \sim D'$. We formally define neighboring datasets below.

**Definition 1** (($\varepsilon, \delta$)-Differential Privacy). *A randomized mechanism $\mathcal{M}$ satisfies ($\varepsilon, \delta$)-DP under neighboring relation $\sim$ if for all $D \sim D'$ and all measurable sets of outputs $Z$ we have*

$$\Pr[\mathcal{M}(D) \in Z] \le e^{\varepsilon} \Pr[\mathcal{M}(D') \in Z] + \delta.$$

In this work, we consider datasets where each datapoint is a single-dimensional real value in the interval $[-1, 1]$. The mechanisms we consider apply more generally to multi-dimensional real-valued queries, and the pitfalls we highlight in this work are present for such input as well. We focus on single-dimensional inputs for simplicity of presentation. Likewise, any mechanism we define on $[-1, 1]$ can be extended to all of $\mathbb{R}$ by clipping to the range $[-1, 1]$. After the appropriate rescaling, this is equivalent to the mechanisms used in practice for tasks including differentially private stochastic gradient descent.

On the domain $[-1, 1]^*$, we define the neighboring definitions of add, remove, and substitution (replacement). We typically want the neighboring relation to be symmetric, which is why add and remove are typically included in a single definition. However, as noted by previous work we need to analyze the add and remove cases separately to get tight results (see, e.g., Zhu et al. (2022)).

**Definition 2** (Neighboring Datasets). *Let $D$ and $D'$ be datasets. If $D'$ can be obtained by adding a datapoint to $D$, then we write $D \sim_A D'$. Likewise, if $D$ can be obtained by adding a datapoint*

*to $D'$, then we write $D \sim_R D'$. Combining these, write $D \sim_{A/R} D'$ if $D \sim_A D'$ or $D \sim_R D'$. Finally, we write $D \sim_S D'$ if $D$ can be obtained from $D'$ by swapping one datapoint for another.*

Note that differential privacy under add and remove implies differential privacy under substitution, with appropriate translation of the privacy parameters. For any pair of neighboring datasets $D \sim_S D'$ that differ in index $i$ we can construct an intermediate dataset $D''$ by removing $D_i$. If $\mathcal{M}$ satisfies $(\varepsilon_A, \delta_A)$-DP and $(\varepsilon_R, \delta_R)$-DP for the add and remove relations, respectively, then $\mathcal{M}$ satisfies $(\varepsilon_A + \varepsilon_R, e^{\varepsilon_A}\delta_R + \delta_A)$-DP under substitution because

$$\Pr[\mathcal{M}(D) \in Z] \le e^{\varepsilon_R}\Pr[\mathcal{M}(D'') \in Z] + \delta_R \le e^{\varepsilon_A + \varepsilon_R}\Pr[\mathcal{M}(D') \in Z] + e^{\varepsilon_A}\delta_R + \delta_A.$$

Definition 1 can be restated in terms of the hockey-stick divergence:

**Definition 3** (Hockey-stick Divergence). *For any $\alpha \ge 0$ the hockey-stick divergence between distributions $P$ and $Q$ is*

$$H_\alpha(P\|Q) := \mathbb{E}_{y \sim Q}\left[\left(\frac{dP}{dQ}(y) - \alpha\right)_+\right] = P(S_\alpha) - \alpha Q(S_\alpha),$$

*where $\frac{dP}{dQ}$ is the Radon–Nikodym derivative and $S_\alpha = \{y | \frac{dP}{dQ}(y) \ge \alpha\}$.*

Specifically, a randomized mechanism $\mathcal{M}$ satisfies $(\varepsilon, \delta)$-differential privacy if and only if $H_{e^\varepsilon}(\mathcal{M}(D)\|\mathcal{M}(D')) \le \delta$ for all pairs of neighboring datasets $D \sim D'$. This restated definition is the basis for the privacy accounting tools we consider in this paper. If we know what choice of neighboring datasets $D \sim D'$ maximizes the expression then we can get optimal parameters by computing $H_{e^\varepsilon}(\mathcal{M}(D)\|\mathcal{M}(D'))$.

The full range of privacy guarantees for a mechanism can be captured by the privacy curve.

**Definition 4** (Privacy Curves). *The privacy curve of a randomized mechanism $\mathcal{M}$ under neighboring relation $\sim$ is the function $\delta_{\mathcal{M}}^{\sim} : \mathbb{R} \to [0, 1]$ given by*

$$\delta_{\mathcal{M}}^{\sim}(\varepsilon) := \min\{\delta \in [0, 1] : \mathcal{M} \text{ is } (\varepsilon, \delta)\text{-}DP\}.$$

*If there is a single pair of neighboring datasets $D \sim D'$ such that $\delta_{\mathcal{M}}^{\sim}(\varepsilon) = H_{e^\varepsilon}(\mathcal{M}(D)\|\mathcal{M}(D'))$ for all $\varepsilon \in \mathbb{R}$, we say that the privacy curve of $\mathcal{M}$ under $\sim$ is realized by the worst-case dataset pair $(D, D')$.*

Unfortunately, a worst-case dataset pair does not always exist. A broader tool that is now frequently used in the computation of privacy curves is the privacy loss distribution (PLD) formalism (Dwork and Rothblum, 2016; Sommer et al., 2019).

**Definition 5** (Privacy Loss Distribution). *Given a mechanism $\mathcal{M}$ and a pair of neighboring datasets $D \sim D'$, the privacy loss distribution of $\mathcal{M}$ with respect to $(D, D')$ is*

$$L_{\mathcal{M}}(D\|D') := \ln(d\mathcal{M}(D)/d\mathcal{M}(D'))_* \mathcal{M}(D),$$

*i.e. $\ln\left(\frac{d\mathcal{M}(D)}{d\mathcal{M}(D')}(y)\right) \sim L_{\mathcal{M}}(D\|D')$ when $y \sim \mathcal{M}(D)$.*

An important caveat is that the privacy loss distribution is defined with respect to a specific pair of datasets, whereas the privacy curve implicitly involves taking a maximum over all neighboring pairs of datasets. Nonetheless, the PLD formalism can be used to recover the hockey-stick divergence via

$$H_{e^\varepsilon}(\mathcal{M}(D)\|\mathcal{M}(D')) = \mathbb{E}_{Y \sim L_{\mathcal{M}}(D\|D')}[1 - e^{\varepsilon - Y}],$$

from which we can reconstruct the privacy curve as

$$\delta_{\mathcal{M}}^{\sim}(\varepsilon) = \max_{D \sim D'}\mathbb{E}_{Y \sim L_{\mathcal{M}}(D\|D')}[1 - e^{\varepsilon - Y}].$$

Lastly, we define the two subsampling procedures we consider in this work: sampling without replacement and Poisson sampling. Given a dataset $D = (x_1, \ldots, x_n)$ and a set $I \subseteq \{1, \ldots, n\}$, we denote by $D|_I := (x_{i_1}, \ldots, x_{i_b})$ the restriction of $D$ to $I = \{i_1, \ldots, i_b\}$.

**Definition 6** (Subsampling). *Let $\mathcal{M}$ take datasets of size[1] $b \geq 1$. The $\binom{n}{b}$-subsampled mechanism $\mathcal{M}_{WOR}$ is defined on datasets of size $n \geq b$ as*

$$\mathcal{M}_{WOR}(D) := \mathcal{M}(D|_I),$$

*where $I$ is a uniform random $b$-subset of $\{1, \ldots, n\}$.*

*On the other hand, given a mechanism $\mathcal{M}$ taking datasets of any size, the $\gamma$-subsampled mechanism $\mathcal{M}_{Poisson}$ is defined on datasets of arbitrary size as*

$$\mathcal{M}_{Poisson}(D) := \mathcal{M}(D|_I),$$

*where $I$ includes each element of $\{1, \ldots, |D|\}$ independently with probability $\gamma$.*

## 3    RELATED WORK

After Dwork and Rothblum (2016) introduced privacy loss distributions, a number of works used the formalism to estimate the privacy curve to arbitrary precision, beginning with Sommer et al. (2019). Koskela et al. (2020; 2021) develop an efficient accountant that efficiently computes the convolution of PLDs by leveraging the fast Fourier transform. Gopi et al. (2021) fine-tunes the application of FFT to speed up the accountant by several orders of magnitude.

The most relevant related paper for our work is by Zhu et al. (2022). They introduce the concept of a dominating pair of distributions. A typical approach to analyzing the privacy guarantees of a mechanism is to consider a pair of datasets that represents the worst-case for the privacy parameters. However, for some problems, determining what is a worst-case pair is not obvious: sometimes there is not a single pair of worst-case datasets. Their notion of a dominating pair of distributions helps formalize the idea of worst-case datasets.

**Definition 7** (Dominating Pair of Distributions (Zhu et al., 2022)). *The ordered pair of distributions $(P, Q)$ dominates a mechanism $\mathcal{M}$ (under some neighboring relation $\sim$) if for all $\alpha \geq 0$*

$$\sup_{D \sim D'} H_\alpha(\mathcal{M}(D)||\mathcal{M}(D')) \leq H_\alpha(P||Q).$$

The hockey-stick divergence of the dominating pair $P$ and $Q$ gives an upper bound on the value $\delta$ for any $\varepsilon$. Note that the distributions $P$ and $Q$ do not need to correspond to distributions of running the mechanism on actual datasets. However, if there exists a pair of neighboring datasets such that $P = \mathcal{M}(D)$ and $Q = \mathcal{M}(D')$ then we can find tight privacy parameters by analyzing the mechanisms with inputs $D$ and $D'$ because $H_{e^\varepsilon}(\mathcal{M}(D)||\mathcal{M}(D'))$ is also a lower bound on $\delta$ for any $\varepsilon$. The definition of dominating pairs of distributions is useful for analyzing the privacy guarantees of composed mechanisms. In this work, we focus on the special case where a mechanism consists of $k$ self-compositions. This is, for example, the case in DP-SGD, in which we run several iterations of the subsampled Gaussian mechanism. The property we need for composition is presented in Theorem 8.

**Theorem 8** (Following Theorem 10 of Zhu et al. (2022)). *If $(P, Q)$ dominates a mechanism $\mathcal{M}$ then $(P^k, Q^k)$ dominates $k$ iterations of $\mathcal{M}$.*

When studying differential privacy parameters in terms of the hockey-stick divergence, we usually focus on the case of $\alpha \geq 1$. Recall that the hockey-stick divergence of order $\alpha$ can be used to bound the value of $\delta$ for an $(\varepsilon, \delta)$-DP mechanism where $\varepsilon = \ln(\alpha)$. We typically do not care about the region of $\alpha < 1$ because it corresponds to negative values of $\varepsilon$. However, it is crucial that definition of dominating pairs of distributions consider these values as well. This is because outputs with negative privacy loss are important for composition and Theorem 8 would not hold if the definition only considered $\alpha \geq 1$. In Sections 5 and 7 we consider distribution where the distributions that bound the hockey-stick divergence for $\alpha \geq 1$ without composition does not bound $\alpha \geq 1$ under composition.

Zhu et al. (2022) studied general mechanisms in terms of dominating pairs of distributions under Poisson subsampling and sampling without replacement. Their work give upper bounds on the privacy parameters based on the dominating pair of distributions of the non-subsampled mechanism. We use some of their results which we introduce later throughout this paper.

---

[1]We treat the sample size and batch size as public knowledge in line with prior work. (Zhu et al., 2022)

## 4 WORST-CASE PAIR OF DATASETS UNDER ADD AND REMOVE RELATIONS

In this section we give pairs of neighboring datasets with provable worst-case privacy parameters under the add and remove neighboring relations separately. We use these datasets as examples of the pitfalls to avoid in the subsequent section, where we discuss the combined add/remove neighboring relation.

**Proposition 9.** *Let $\mathcal{M}(x_1, \ldots, x_n) := \sum_{i=1}^{n} x_i + N(0, \sigma^2)$ denote the Gaussian mechanism.*

1. *There exists a dataset pair $D \sim_R D'$ such that $\delta_{\mathcal{M}_{Poisson}}^{\sim_R}$ is realized by $(D, D')$ and $\delta_{\mathcal{M}_{Poisson}}^{\sim_A}$ is realized by $(D', D)$ (recall Definition 4).*

2. *Likewise, there exists a dataset pair $D \sim_R D'$ such that $\delta_{\mathcal{M}_{WOR}}^{\sim_R}$ is realized by $(D, D')$ and $\delta_{\mathcal{M}_{WOR}}^{\sim_A}$ is realized by $(D', D)$.*

Our worst-case datasets can be found by reduction to one of the main results of Zhu et al. (2022).

**Theorem 10** (Theorem 11 of Zhu et al. (2022))**.** *Let $\mathcal{M}$ be a randomized mechanism, let $\mathcal{M}_{Poisson}$ be the $\gamma$-subsampled version of the mechanism, and let $\mathcal{M}_{WOR}$ be the $\binom{n}{b}$-subsampled version of the mechanism on datasets of size $n$ and $n-1$ with $\gamma = b/n$.*

1. *If $(P, Q)$ dominates $\mathcal{M}$ for add neighbors then $(P, (1-\gamma)P + Q)$ dominates $\mathcal{M}_{Poisson}$ for add neighbors and $((1-\gamma)P + Q, P)$ dominates $\mathcal{M}_{Poisson}$ for removal neighbors.*

2. *If $(P, Q)$ dominates $\mathcal{M}$ for substitute neighbors then $(P, (1-\gamma)P + Q)$ dominates $\mathcal{M}_{WOR}$ for add neighbors and $((1-\gamma)P + Q, P)$ dominates $\mathcal{M}_{WOR}$ for removal neighbors.*

We sketch below the proof of Proposition 9. We also note that the proposition can also be proved for the Laplace mechanism by an identical argument where noise from the Laplace distribution replaces Gaussian noise.

By symmetry, we will focus the proof sketch on the add neighboring relation. Now, from Theorem 10 we know that $(\mathcal{N}(0, \sigma^2), (1 - \gamma)\mathcal{N}(0, \sigma^2) + \gamma\mathcal{N}(1, \sigma^2))$ dominates $\mathcal{M}_{Poisson}$ and $(\mathcal{N}(0, \sigma^2), (1 - \gamma)\mathcal{N}(0, \sigma^2) + \gamma\mathcal{N}(2, \sigma^2))$ dominates $\mathcal{M}_{WOR}$. We know that the dominating pair is tight if there exists a pair of neighboring datasets for which the hockey-stick divergence matches the dominating pair for all $\alpha$. We can easily show that such inputs exist. Say that $D_i'$ is the element in $D'$ that is not in $D$. The case of Poisson subsampling is simple. We set all elements in $D$ to 0 and set $D_i' = 1$. Then $\mathcal{M}_{Poisson}(D')$ is centered around 1 if $D_i'$ is included in the batch and 0 otherwise. That is, the distribution matches the dominating pair.

The worst-case datasets are similar although slightly different for $\mathcal{M}_{WOR}$. We still set $D_i' = 1$ but now we set all elements of $D$ to $-1$. For a batch of $\mathcal{M}_{WOR}(D')$ consisting of $m = \gamma n$ elements the output is either centered around $-m$ or $-m + 2$ depending on whether of not $D_i'$ is included in the batch whereas $\mathcal{M}_{WOR}(D) = \mathcal{N}(-m, \sigma^2)$. Shifting both distributions by $m$ gives us the dominating pair from Theorem 10 and this does not affect the hockey-stick divergence.

As such, there exists true worst-case datasets for both sampling schemes under the add and remove relations, respectively. Crucially, the distributions above show us that under the add and remove relations we must add noise with twice the magnitude when sampling without replacement compared to Poisson subsampling! The intuition for this difference is that the subroutine behaves similarly to the add/remove neighboring relation when using Poisson subsampling, whereas it resembles the substitution neighborhood when sampling without replacement. When $D_i'$ is included in the batch another datapoint is 'pushed out' of the batch when sampling without replacement. Due to this parallel one might hope that the difference in privacy parameters between Poisson subsampling and sampling without replacement only differ by a small constant similar to the difference between the add/remove and substitution neighboring relations. That is indeed the case for many parameters, but as we show in Section 7 this assumption unfortunately does not always hold.

## 5 NO WORST-CASE PAIR OF DATASETS UNDER ADD/REMOVE RELATION

So far, we have considered privacy curves defined for all $\varepsilon \in \mathbb{R}$, which is a necessary subtlety for PLD privacy accounting tools (e.g., Theorem 8). In this section, we relax the strong notion of worst-

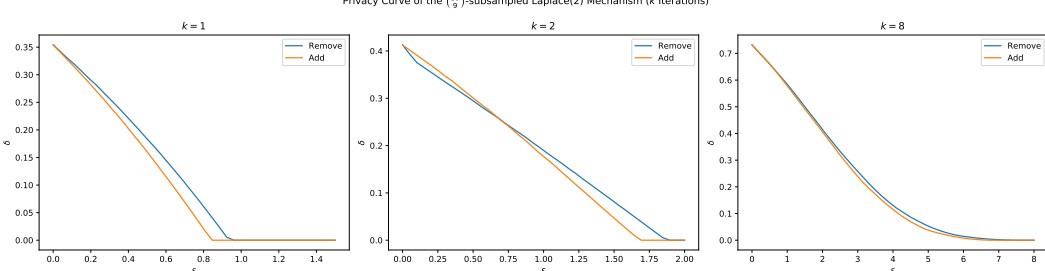

Figure 1: The privacy curves for the composed, subsampled Laplace mechanism under the remove and add neighboring relations respectively.

case dataset from Definition 4 to the more familiar setting where $\varepsilon \geq 0$. Our main result is to give an example of a mechanism that admits a worst-case dataset under $\sim_{A/R}$ and $\varepsilon \geq 0$ when run for a single iteration, yet fails to admit a worst-case dataset (for $\varepsilon \geq 0$) when run for multiple iterations. This violates an implicit assumption made by some privacy accountants. We can correct for this by computing the privacy curve under $\sim_A$ and $\sim_R$ separately and then taking the maximum.

**Proposition 11.** *Consider the Laplace mechanism $\mathcal{M}(x_1, \ldots, x_n) := \sum_{i=1}^n x_i + Lap(0, s)$ and let $\mathcal{M}_{WOR}$ be the $\binom{n}{b}$-subsampled mechanism. Then, for some choice of $k > 1$ and $b \geq 1$, there is no dataset-pair $D \sim_{A/R} D'$ such that*

$$\delta^{\sim_{A/R}}_{\mathcal{M}_{WOR}^k}(\varepsilon) = H_{e^\varepsilon}(\mathcal{M}_{WOR}^k(D) \| \mathcal{M}_{WOR}^k(D'))$$

*for all $\varepsilon \geq 0$.*

Note that this result can be extended easily to the $\gamma$-subsampled mechanism $\mathcal{M}_{Poisson}$ as well. In any case, as we argued in Section 4, the privacy curves of $\mathcal{M}_{WOR}$ under both add and remove neighbouring relations are realized by worst-case datasets. That is, we can find datasets $D \sim_R D'$ such that $\delta^{\sim_R}_{\mathcal{M}_{WOR}}$ is realized by $(D, D')$ and $\delta^{\sim_A}_{\mathcal{M}_{WOR}}$ is realized by $(D', D)$. Moreover, it is generally the case that the privacy curve of a subsampled mechanism under $\sim_R$ dominates the same privacy curve under $\sim_A$ when $\varepsilon \geq 0$ (see e.g. Proposition 30 of Zhu et al. (2022) or Theorem 5 of Mironov et al. (2019)). In our case, the subsampled Laplace mechanism satisfies the symmetry conditions of Zhu et al. (2022) Proposition 30, we also have that

$$\delta^{\sim_{A/R}}_{\mathcal{M}_{WOR}}(\varepsilon) = \delta^{\sim_R}_{\mathcal{M}_{WOR}}(\varepsilon) \geq \delta^{\sim_A}_{\mathcal{M}_{WOR}}(\varepsilon)$$

for $\varepsilon \geq 0$.

Now, noting that $\delta^{\sim_{A/R}}_{\mathcal{M}_{WOR}}(\varepsilon) = \max\{\delta^{\sim_A}_{\mathcal{M}_{WOR}}(\varepsilon), \delta^{\sim_R}_{\mathcal{M}_{WOR}}(\varepsilon)\}$, Proposition 11 can be proved by plot. Figure 1 shows several variations of the curves $\delta^{\sim_A}_{\mathcal{M}_{WOR}^k}$ and $\delta^{\sim_R}_{\mathcal{M}_{WOR}^k}$, which have been computed numerically by Monte Carlo simulation (as in e.g. Wang et al. (2023)). These curves are seen to cross in the region $\varepsilon \geq 0$ when $k = 2$.

Interestingly, the phenomenon emerges when $k$ increases from 1 but disappears again for larger values of $k$. This is better understood by considering the privacy loss distributions $L_{\mathcal{M}_{WOR}}(D \| D')$ and $L_{\mathcal{M}_{WOR}}(D' \| D)$, whose CDFs are shown in Figure 2.

The add relation PLD $L_{\mathcal{M}_{WOR}}(D' \| D)$, which has a large mass at $-\ln(0.1 + 0.9/e)$, is more likely to take on positive values compared to the remove relation PLD $L_{\mathcal{M}_{WOR}}(D \| D')$, which tends to take on larger values due to its mass being slightly more spread out over $[0, \infty)$. This effect is amplified by composition, which leads to the composed add PLD $L_{\mathcal{M}_{WOR}^2}(D \| D')$ being much more likely to take on small positive values compared to $L_{\mathcal{M}_{WOR}^2}(D' \| D)$. This in turn causes the privacy curve for $\mathcal{M}_{WOR}^2$ to be larger under the add relation compared to the remove relation when $\varepsilon$ is small.

Under a larger number of compositions, however, it is known that both PLDs converge to a Gaussian distribution (Dong et al., 2019). This explains why $\delta^{\sim_R}_{\mathcal{M}_{WOR}^k}$ and $\delta^{\sim_A}_{\mathcal{M}_{WOR}^k}$ tend toward one another and the crossing behavior appears to vanish.

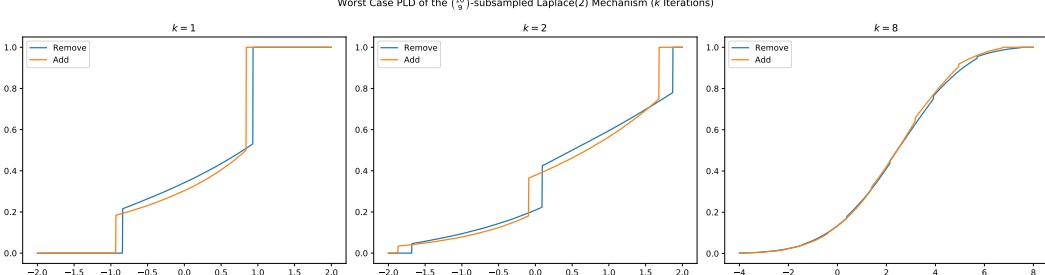

Figure 2: The PLDs of the subsampled Laplace mechanism are mixed, i.e. partially discrete and partially continuous. The discontinuities of the CDF indicate the atoms of the discrete component of each distribution.

## 6 PRIVACY GUARANTEE DIFFERENCE BETWEEN SAMPLING SCHEMES

In this section we explore settings where the privacy parameters between Poisson subsampling and sampling without replacement differ significantly. We focus on the subsampled Gaussian mechanism since this is the mechanism of choice for DP-SGD. One approach for choosing privacy-specific hyperparameters is to fix $\delta$ and the number of iterations to run DP-SGD. We can then compute the minimum value of the noise multiplier $\sigma$ required to achieve $(\varepsilon, \delta)$-DP for various sampling rates. We follow this approach for our example. We fix $\delta = 10^{-6}$ and the number of iterations to $10,000$. We then vary the sampling rate between $10^{-4}$ to $1$ and use the *PLD* accountant implemented in the Opacus library (Yousefpour et al., 2021) to compute $\sigma$.

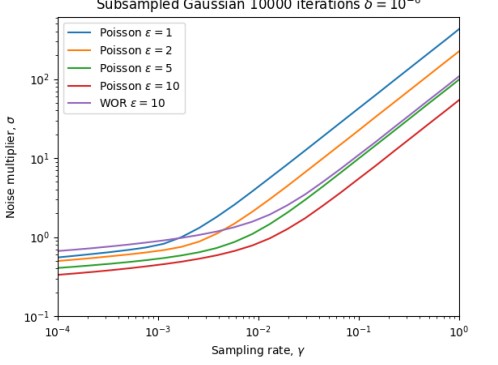

| $\delta$ | $\varepsilon$ (Poisson) | $\varepsilon$ (WOR) |
|---|---|---|
| $10^{-7}$ | 1.19 | 17.48 |
| $10^{-6}$ | 0.96 | 15.26 |
| $10^{-5}$ | 0.80 | 12.98 |
| $10^{-4}$ | 0.64 | 10.62 |

Figure 3: Plot of smallest noise multiplier $\sigma$ required to achieve certain privacy parameters for the subsampled Gaussian mechanism with varying sampling rates under add/remove. Each line shows a specific value of $\varepsilon$ for either Poisson subsampling or sampling without replacement. The parameter $\delta$ is fixed to $10^{-6}$ for all lines. The table shows the difference in privacy parameters for sampling rate $\gamma = 0.001$ and noise multiplier $\sigma = 0.8$ for multiple values of $\delta$.

In Figure 3 we plot the noise multiplier required to achieve $(\varepsilon, \delta)$-DP with Poisson subsampling for $\varepsilon \in \{1, 2, 5, 10\}$. For comparison we plot the noise multiplier required to achieve $(10, \delta)$-DP when sampling without replacement. From the previous section we know that this is twice the noise required for $(10, \delta)$-DP with Poisson subsampling. The plot is clearly divided into two regions. When the sampling rate is high, the noise multiplier scales linearly in the sampling rate. However, for sufficiently low sampling rates the noise multiplier decreases much slower.

This effect has been observed previously (see Ponomareva et al. (2023); Anil et al. (2022) for similar plots). Ideally, we want to choose parameters on the right side of this hinge as the benefits of privacy amplification wear off on the left side. For this reason some work has observed improved results when using large batch size (see, e.g., Dörmann et al. (2021)).

However, one might choose to select parameters close to the hinge. This can be problematic if the accountant assumes Poisson sampling but sampling without replacement is used. The hinge happens when $\sigma$ is slightly less than 1 for Poisson sampling and therefore it happens when it is slightly less than 2 for sampling without replacement. The consequence can be seen for the curve for sampling without replacement in Figure 3. For large sampling rates the noise requires roughly matches that for $\varepsilon = 5$ with Poisson subsampling. But for the small sampling rates we have to add more noise than for $\varepsilon = 1$ with Poisson subsampling. As such, if we use an accountant for Poisson subsampling with a target of $\varepsilon = 1$ but our implementation using sampling without replacement the actual value of $\varepsilon$ could be above 10! We could hope that this increase would be offset if we allow for some slack in $\delta$ as well. However, as seen in the table of Figure 3 there can still be a big gap in $\varepsilon$ even when we allow several orders of magnitude difference in $\delta$.

## 7 SUBSTITUTION

In this section we consider the substitution neighboring relation. Unfortunately, finding tight bounds under the substitution relation can be challenging. We have already shown how the worst-case pair of datasets can differ depending on the value of $\varepsilon$ under the add/remove relation. However, the solution to that problem is straightforward because we simply find the parameters under the add and remove relations separately and report the maximum value. Here we show that such a solution is not always possible under the substitution relation using a small concrete example for sampling without replacement.

First, we restate another result from Zhu et al. (2022) which we use throughout the section. Similar to the case of add and remove, they show that without composition the worst-case datasets depend on whether or not $\alpha$ is above 1.

**Theorem 12** (Proposition 30 of Zhu et al. (2022))**.** *If* $(P, Q)$ *dominates* $\mathcal{M}$ *under the substitution neighborhood relation for datasets of size* $\gamma n$ *then under the substitution neighborhood for datasets of size* $n$ *we have*

$$\delta(\alpha) \leq \begin{cases} H_\alpha((1-\gamma)P + \gamma Q || P) & \text{if } \alpha \geq 1; \\ H_\alpha(P || (1-\gamma)P + \gamma Q) & \text{if } 0 < \alpha < 1, \end{cases}$$

*where* $\delta(\alpha)$ *is the largest hockey-stick divergence of order* $\alpha$ *between running* $\mathcal{M}_{WOR}$ *for a pair of neighboring datasets.*

Next we address a mistake made in related work. In their work on computing tight differential privacy guarantees Koskela et al. (2020) considered worst-case neighboring datasets for the subsampling Gaussian mechanism under multiple sampling techniques and neighboring relations. In the case of substitution they compute the hockey-stick divergence between the pair of distributions $(1-\gamma)\mathcal{N}(0, \sigma^2) + \gamma\mathcal{N}(-1, \sigma^2)$ and $(1-\gamma)\mathcal{N}(0, \sigma^2) + \gamma\mathcal{N}(1, \sigma^2)$. They consider the same distributions under both Poisson subsampling and sampling without replacement and as such make the claim that the privacy curve is identical between the two schemes under the substitution relation. Unfortunately, this is not true and it is also contradicted by our example from Section 6. If this claim was true the privacy curves for the two schemes would never differ significantly for add/remove either since the privacy guarantees of substitution never better than under add/remove for the mechanism.

We see that their distributions do not follow the structure of Theorem 12. They compare the datasets $D$ and $D'$ where $D_i = -1$, $D'_i = 1$ and all other entries are 0. This is an understandable mistake as it seems the intuition behind this choice is that we do not want any additional randomness from other elements than the $i$th entry. For Poisson subsampling this requires the elements to have a value of 0, but when sampling without replacement this holds for any pairs of datasets where all other elements have the same value as we sample a fixed amount.

Instead we can consider the datasets where $D_i = 1$, $D'_i = -1$ and all other elements are $-1$. It follows from Theorem 12 that these are worst-case neighbouring datasets without composition. However, since this is not a dominating pairs of distributions it is more complicated when we consider composition. One might hope that we could simply compute the hockey-stick divergence of the self-composed distributions in both directions similar to the add/remove case. But sometimes that is not sufficient because we can combine the directions unlike with the add and remove cases. Next we give a minimal counterexample to showcase this challenge.

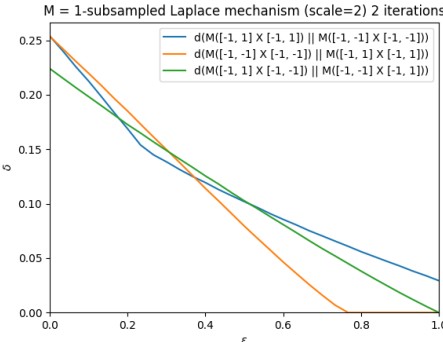

Figure 4: Hockey-stick divergence of the Laplace mechanism when sampling without replacement under substitution neighborhood. The worst-case pair of datasets depends on the value of $\varepsilon$.

Figure 4 shows the hockey-stick divergence as a function of $\varepsilon$ for three pairs of neighboring datasets. We consider datasets of size 2 where we sample batches with a single element that is $\gamma = 0.5$. We apply the Laplace mechanism with a scale of 2. We compose the subsampled mechanism with itself only once, as in we run 2 queries. In all cases the neighboring datasets agree on index 1 and differ in index 2. For the first line $D_1 = D_1' = -1$, $D_2 = 1$, and $D_2' = -1$ for both queries. The second line switches $D$ and $D'$ compared to the first line. So far this is very similar to the add and remove cases, the interesting case is the third line. We still have that $D_1 = D_1' = -1$ for both queries but now the second index differs across queries as we have $D_2 = 1$ and $D_2' = -1$ for the first query, but $D_2 = -1$ and $D_2' = 1$ for the second query. This pair of datasets matches the datasets from the first line in one query and the second line in the other. We can see from Figure 4 that the worst pair of datasets depends on the value of $\varepsilon$ with all three pairs being the worst within one range.

The above example is possible because the distributions for a query can match either the worst-case for add or remove. And since queries can be independent we can match the add case for one query and remove case for the other. This scales up with the number of compositions. For $k$ compositions, we have to consider $k + 1$ ways of dividing up the queries between matching the add or the remove case. This significantly slows down the accountants in contrast to the 2 cases for add/remove. Worse still, we do not have a formal proof that one of $k + 1$ cases is the worst-case pair of datasets.

## 8    DISCUSSION

We have highlighted two issues that arise in the practice of privacy accounting.

First, we have given a concrete example where the worst-case dataset (for $\varepsilon \geq 0$) of a subsampled mechanism fails to be a worst-case dataset once that mechanism is composed. Care should therefore be taken to ensure that the privacy accountant computes privacy guarantees with respect to a true worst-case dataset for a given choice of $\varepsilon$.

Secondly, we have shown that the privacy parameters for a subsampled and composed mechanism can differ significantly for different subsampling schemes. This can be problematic if the privacy accountant is assuming a different subsampling procedure from the one actually employed. We have shown this in the case of Poisson sampling and sampling without replacement but the phenomenon is likely to occur when comparing Poisson sampling to shuffling as well. Computing tight privacy guarantees for the shuffled Gaussian mechanism remains an important open problem. It is best practice to ensure that the implemented subsampling method matches the accounting method. When this is not practical, the discrepancy should be disclosed.

We conclude with two recommendations for practitioners applying privacy accounting in the DP-SGD setting. We recommend disclosing the privacy accounting hyperparameters for the sake of reproducibility (see Section 5.3.3 of Ponomareva et al. (2023) for a list of suggestions). Finally, we also recommend that, when comparisons are made between DP-SGD mechanisms, the privacy accounting for both should be re-run for the sake of fairness.

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
