# OpenReview forum: "Avoiding Pitfalls for Privacy Accounting of Subsampled Mechanisms under Composition"
_ICLR.cc/2024/Conference — Submitted to ICLR 2024_

### Official Review · Reviewer_G1qb · 2023-10-31

**Soundness:** 2 fair
**Presentation:** 3 good
**Contribution:** 2 fair
**Rating:** 3
**Confidence:** 4

**Summary:**

The paper addresses potential pitfalls when using so called numerical privacy accounting to compute the $(\varepsilon,\delta)$-guarantees for compositions of DP algorithms. Especially, the paper focuses on two points (citations from the paper):

a) "some privacy accountants assume that the privacy guarantees for the composition of a subsampled mechanism are determined by self-composing the worst-case datasets for the uncomposed mechanism"

and

b) "Poisson subsampling is sometimes assumed to have similar privacy guarantees compared to sampling without replacement."


Some background: Using the Rényi divergence has been a popular way of analysing compositions of DP mechanisms since the works of Abadi et al. (2016) and Mironov (2017), especially due to the tightness of the RDP accounting compared to purely analytical approaches. So called numerical accounting and was proposed in works by Sommer et al. (2019), Koskela et al. (2020) and Gopi et al. (2021). It directly approximates the hockey-stick divergence and often gives tighter bounds than the RDP approach which incurs a small loss when converting the RDP parameters to $(\varepsilon,\delta)$-bounds. Zhu et al. (AISTATS 2022, https://proceedings.mlr.press/v151/zhu22c/zhu22c.pdf) set the hockey-stick divergence based methods on a more rigorous footing by introducing the concept of dominating pairs of distributions. This also allows obtaining rigorous $(\varepsilon,\delta)$-bounds for adaptive compositions of DP mechanisms.

**Strengths:**

Generally well-written paper and a timely topic. The auditing methods are getting all the time more accurate at estimating the $\varepsilon$-values, so it would be good to get the accurate computing of the formal guarantees right. It seems there are still no clear subsampling amplification result in the literature for, e.g., carrying out accounting using the dominating pairs of distributions in case of substitution neighbourhood relation of datasets (related to the “second pitfall”), so it makes sense to consider this problem.

**Weaknesses:**

All in all, I think the contribution of the paper is very limited. And I think the paper is outdated in a sense that neither of the mentioned issues are actually pitfalls in numerical accounting (explained below).

About the pitfalls:

The point a) was addressed by the work of Zhu et al. (2022) which considers the dominating pairs of distributions. As the authors point out, generally there are no neighboring datasets $X$ and $Y$ such that the $(\varepsilon,\delta)$-bound for pairs of outcomes $\big(\mathcal{M}(X),\mathcal{M}(Y)\big)$ would be $(\varepsilon,\delta)$-bounds for all neighboring datasets. Outside of the mechanisms that use additive Gaussian noise, there are numerous such examples. Consider, e.g., the exponential mechanism, for which obtaining accurate $(\varepsilon,\delta)$-bounds is very tedious:

Dong, J., Durfee, D., & Rogers, R. (2020, November). Optimal differential privacy composition for exponential mechanisms. In International Conference on Machine Learning (pp. 2597-2606). PMLR.

The issue that you don't have a worst-case pair of datasets is exactly what the definition of the dominating pairs of distribution addresses. Also, Zhu et al. (2022) also show that a tightly dominating pair distributions always exists for a given mechanism.

For the pitfall b) I agree that the claim that the pair of distributions $P = q \cdot \mathcal{N}(1,\sigma^2) + (1-q) \cdot \mathcal{N}(0,\sigma^2)$, $Q = q \cdot \mathcal{N}(-1,\sigma^2) + (1-q) \cdot \mathcal{N}(0,\sigma^2)$ gives a dominating pair of distributions in case of subsampling without replacement and substitute neighbourhood relation of datasets is not correct. I believe it is true for the Poisson subsampling in case of substitute neighbourhood relation of datasets, I think you can show this as in case of the Rényi divergence and use the analysis for the Poisson subsampling and add/remove neighborhood relation of datasets, which is given in

Mironov, I., Talwar, K., & Zhang, L. (2019). R\'enyi differential privacy of the sampled gaussian mechanism. arXiv preprint arXiv:1908.10530.

The correct bound for hockey-stick divergence in the case of subsampling without replacement and substitute neighbourhood relation of datasets is given in Proposition 30 by Zhu et al. (2022). So, as far as I see, this problem is solved, and in the most well-known software libraries that use numerical accounting and dominating pairs of distributions, namely the "autodp" by Wang et al. and "PRV accountant" by Gopi et al., Opacus and Google DP library, correct formulas are used.

I think that the claim that "Poisson subsampling is sometimes assumed to have similar privacy guarantees compared to sampling without replacement" is not true in that Proposition 30 by Zhu et al. (2022) gives bounds for both. And they are of the same form, but in of them the pair $(P,Q)$ is a dominating pair under the add/remove relation and in the other one under the substitute relation, so it is clear that the latter leads to higher $\varepsilon$-values. The bounds are in two parts and one can determine a numerical dominating pair of distributions using, e.g, methods by

Doroshenko, V., Ghazi, B., Kamath, P., Kumar, R., & Manurangsi, P. (2022). Connect the Dots: Tighter Discrete Approximations of Privacy Loss Distributions. Proceedings on Privacy Enhancing Technologies, 4, 552-570.

**Questions:**

- What is special about the hockey stick divergence when thinking about the worst-case pairs of datasets and worst-case pairs of distributions? All the potential problems w.r.t. to finding the worst-case pair of distributions for the hockey-stick divergence would be problems for RDP accounting (or when using other $f$-divergences than the $\alpha$-divergence) as well, right? Commonly the worst-case pairs of distributions are 1-dimensional and can be seen as some sort of general post-processing of the outcomes from neighboring datasets, and then the worst-case distributions for the hockey-stick divergence would similarly be worst-case distributions for other $f$-divergences and for the Rényi divergence as they satisfy the data-processing inequality.

- Comment: it would be interesting to see some new results related to this topic, e.g., on how would the analytically expressed pair of dominating distributions look like under the subsampling amplification (in all cases) as the bounds of Proposition 30 by Zhu et al. (2022) (which you also cite) are given in two parts.

---

> ### Author Response · Authors · 2023-11-18
>
> Thank you for your thorough review.
>
> > The point a) was addressed by the work of Zhu et al. (2022)...
>
> The intent of our example was not to show that some mechanisms do not have worst-case datasets. As you point out there are many such mechanisms. Our example with the Laplace mechanism simply shows that even a simple mechanism that has a worst-case pair of datasets might not have a worst-case pair of datasets under subsampling and composition. We will make this point more explicit in the paper. As for whether or not this is still a pitfall for privacy accountants please see our response below.
>
> > The correct bound for hockey-stick divergence in the case of subsampling without replacement and substitute neighbourhood relation of datasets is given in Proposition 30 by Zhu et al. (2022). So, as far as I see, this problem is solved, and in the most well-known software libraries that use numerical accounting and dominating pairs of distributions, namely the "autodp" by Wang et al. and "PRV accountant" by Gopi et al., Opacus and Google DP library, correct formulas are used.
>
> Proposition 30 gives the upper bound of the hockey-stick divergence for a single iteration. However, it is not a dominating pair of distributions since the order of the pair differs between alpha below or above 1. In Section 7 we show that the pair of datasets considered in Proposition 30 is not sufficient to give tight bounds under composition for general mechanisms. You are correct that Zhu et al. give a construction of a dominating pair of distributions in Corollary 32. However, there is no implementation of this construction to the best of our knowledge.
> Regarding implementations of accountants for the subsampled Gaussian mechanism: the “PRV accountant”/Opacus (and PLD Accountant) only computes the “remove” relation. The “autodp” and Google DP library correctly computes both the “add” and “remove” relations and takes the point-wise maximum, as in Theorem 11 of Zhu et al. As mentioned in our response to Reviewer LTUo we conjecture that in the case of the Gaussian mechanism, it is sufficient to consider the “remove” datasets for subsampling without replacement under the substitution relation. However, we are not aware of any proof and therefore we disagree with the claim that this problem is solved.
>
> > ...the claim that "Poisson subsampling is sometimes assumed to have similar privacy guarantees compared to sampling without replacement" is not true…
>
> We agree that this claim is too strong as is and should be rephrased. The privacy parameters are not assumed to be identical. However, the privacy parameters are sometimes assumed to be close as is the case between the add/remove and substitution neighboring relations. We base this claim on the fact that accountants for Poisson subsampling are often used when it is not the implemented sampling scheme.
>
> > What is special about the hockey stick divergence…?
>
> The RDP accountant computes the RDP guarantees of a single iteration for a given order and then applies the RDP composition theorem. Due to the privacy scaling under the RDP composition theorem, it is sufficient to find the worst-case datasets for only a single iteration when using the RDP accountant. One could use a similar approach for hockey-stick divergence by computing a single (ε, δ)-DP guarantee and applying the advanced composition theorem. However, (ε, δ)-DP guarantees do not compose as well as the RDP guarantee. Therefore, the problem of determining the worst-case datasets under composition only seems to arise for exact accountants.

---

> ### Comment · Reviewer_G1qb · 2023-11-18
>
> Thank you for the reply.
>
> > Our example with the Laplace mechanism simply shows that even a simple mechanism that has a worst-case pair of datasets might not have a worst-case pair of datasets under subsampling and composition.
>
> I still don't fully understand why would one want to find the worst-case pair of datasets. I think it is clear that one has to consider a bound that holds for all pairs of datasets.
>
> > You are correct that Zhu et al. give a construction of a dominating pair of distributions in Corollary 32. However, there is no implementation of this construction to the best of our knowledge.
>
> Such a construction can also be found in "Doroshenko et al. (2022). Connect the Dots: Tighter Discrete Approximations of Privacy Loss Distributions. Proceedings on Privacy Enhancing Technologies." and it is a fairly simple algorithm. But I agree this should be addressed more thoroughly.
>
> > Regarding implementations of accountants for the subsampled Gaussian mechanism: the “PRV accountant”/Opacus (and PLD Accountant) only computes the “remove” relation. The “autodp” and Google DP library correctly computes both the “add” and “remove” relations and takes the point-wise maximum, as in Theorem 11 of Zhu et al.
>
> That is an interesting observation (and perhaps someone should tell the authors), but I don't think these findings make a sufficient contribution for a paper.
>
> > As mentioned in our response to Reviewer LTUo we conjecture that in the case of the Gaussian mechanism, it is sufficient to consider the “remove” datasets for subsampling without replacement under the substitution relation.
>
> I think proving this (even if it holds only for the Gaussian mechanism) would strengthen the paper.
>
> > The privacy parameters are not assumed to be identical. However, the privacy parameters are sometimes assumed to be close as is the case between the add/remove and substitution neighboring relations. We base this claim on the fact that accountants for Poisson subsampling are often used when it is not the implemented sampling scheme.
>
> It indeed seems to be the case that the accountants for Poisson subsampling (with remove/add neighbourhood relation) are often used when they shouldn't be. And I think the same often happens in case of RDP accountants. Anyhow, I think the differences between neighbourhood relations and sampling schemes are correctly reflected in the results of Zhu et al. (2022). I think it would be interesting to see what are (as tight as possible) dominating pairs (either numerical or analytical) in various cases.
>
> > The RDP accountant computes the RDP guarantees of a single iteration for a given order and then applies the RDP composition theorem.
>
> As far as I understand, the RDP accountants commonly compute the RDP guarantees for several orders and then convert the RDP-epsilons of different orders to approximate DP-guarantees using some conversion rule.
>
> > Due to the privacy scaling under the RDP composition theorem, it is sufficient to find the worst-case datasets for only a single iteration when using the RDP accountant. One could use a similar approach for hockey-stick divergence by computing a single (ε, δ)-DP guarantee and applying the advanced composition theorem. However, (ε, δ)-DP guarantees do not compose as well as the RDP guarantee. Therefore, the problem of determining the worst-case datasets under composition only seems to arise for exact accountants.
>
> I don't exactly follow here. If we want to have an accurate RDP accountant, we would need to have RDP bounds for several RDP orders (the conversion in the end).
>
> All in all I think the topic is actual and the subtleties of hockey stick vs. Rényi divergence based accountants haven't been fully addressed, but this paper does not really propose any solutions and also I think the criticism is not sharp enough to make for a paper.

---

> ### Comment · Reviewer_G1qb · 2023-11-20
>
> Still getting back to this:
>
> > Due to the privacy scaling under the RDP composition theorem, it is sufficient to find the worst-case datasets for only a single iteration when using the RDP accountant. One could use a similar approach for hockey-stick divergence by computing a single (ε, δ)-DP guarantee and applying the advanced composition theorem. However, (ε, δ)-DP guarantees do not compose as well as the RDP guarantee. Therefore, the problem of determining the worst-case datasets under composition only seems to arise for exact accountants.
>
> I believe that different RDP orders could have different worst-case pairs of datasets, similarly as you might not have a worst-case pair for the hockey-stick divergence. To get the benefit out of RDP accounting, one should keep track of several RDP orders. Often the dominating pairs of distributions are obtained by some post-processing (consider e.g. additive Laplace or Gaussian noise), and the resulting pair is a dominating pair for both Rényi and HS divergence, and often this pair is even tight in sense that it exactly gives the divergence of the mechanism evaluated on different dataset pairs (e.g., Gaussian mechanism). If we don't have a worst-case pair of datasets (consider, e.g., the exponential mechanism) but want to find tight bounds for a given divergence, I think one encounters difficulties both with Rényi and HS divergence based accounting.
>
> (By the way, I would rather use the term numerical accounting than exact accounting when talking about the HS divergence based approach.)
>
> > even a simple mechanism that has a worst-case pair of datasets might not have a worst-case pair of datasets under subsampling and composition.
>
> Thinking about this, I believe that the same pair of datasets would be the worst-case pair for compositions as well as for a single call of the mechanism (at least for non-adaptive compositions, I don't think we can expect to find worst-case pairs of datasets for adaptive compositions in general). For subsampling, again, the possible difficulties when trying to find the worst-case dataset pairs would be shared by other accounting methods such as the RDP, I think.
>
> I also share the criticism with reviewer dGjF in that Figure 1 should not be used as a proof (Proposition 11). I also think that the example with Laplace noise is a bit difficult to follow, some more formal derivation e.g. in the appendix would have helped (I suppose that could be even dealt analytically).
>
> I think the paper has some interesting points, but still needs more work. Some constructive results would definitely strengthen the paper.

---

> > ### Author Response · Authors · 2023-11-22
> >
> > > I still don't fully understand why would one want to find the worst-case pair of datasets. I think it is clear that one has to consider a bound that holds for all pairs of datasets.
> >
> > The privacy parameters are determined as a maximum over all pairs of neighboring datasets by definition. Each pair of datasets gives us a lower bound. From Theorem 11 of Zhu et al. we know that in the case of add and remove the privacy parameters are upper bounded by the hockey-stick divergence between two distributions. These distributions correspond to running the subsampling mechanism with an actual pair of neighboring datasets (Assuming there exist inputs that correspond to a dominating pair of distributions for the mechanism without sampling). Their result is significant because it implies that the privacy curve is described exactly by the hockey-stick divergence between those distributions. As such, the error of the privacy parameter estimate is determined only by the error of the estimate of the hockey-stick divergence for the numerical accountants. This is in contrast to for example the RDP accountant which gives us an upper bound but no guarantee on how tight the estimate is. Tight privacy accounting is important when comparing the performance of different mechanisms with the same privacy guarantees.
> >
> > Our counter-example in Section 7 is a negative result showing that no equivalent statement to Theorem 11 exists for arbitrary mechanisms under the substitution neighborhood for subsampling without replacement.
> >
> > > Such a construction can also be found in "Doroshenko et al. (2022). Connect the Dots: Tighter Discrete Approximations of Privacy Loss Distributions. Proceedings on Privacy Enhancing Technologies." and it is a fairly simple algorithm. But I agree this should be addressed more thoroughly.
> >
> > For the experiments in their paper, they likely compute the bound for add and remove separately since they use the Google DP library. However, it seems that their construction should indeed work when combined with the bounds by Zhu et al. (Proposition 30). Thank you for pointing that out. We had not realized that. It is not clear to us how lossy this approach is under composition since the PLD would not correspond to an actual pair of datasets.
> >
> > > As far as I understand, the RDP accountants commonly compute the RDP guarantees for several orders and then convert the RDP-epsilons of different orders to approximate DP-guarantees using some conversion rule. + I don't exactly follow here. If we want to have an accurate RDP accountant, we would need to have RDP bounds for several RDP orders (the conversion in the end). + I believe that different RDP orders could have different worst-case pairs of datasets, similarly as you might not have a worst-case pair for the hockey-stick divergence. To get the benefit out of RDP accounting, one should keep track of several RDP orders.
> >
> > Yes, the RDP accountant typically computes many orders and reports the minimum parameters after conversion. We could use the same idea to compute several $(\varepsilon, \delta)$-DP guarantees of a single iteration and report the minimum result from applying the advanced composition theorem. But that would still be a bad approach for hockey-stick divergence which is the reason other techniques are used.
> >
> > > Thinking about this, I believe that the same pair of datasets would be the worst-case pair for compositions as well as for a single call of the mechanism
> >
> > This will often be the case but it is only guaranteed if running the mechanism on the pair of neighboring datasets corresponds to a dominating pair of distributions. We should have been more precise in our response that this is what we meant by a mechanism that has a worst-case pair of datasets. Otherwise, we could construct a mechanism that has the same distribution as the subsampled Laplace mechanism. Then the pair of datasets that has the largest hockey-stick divergence for any $\alpha \geq 1$ ($\varepsilon \geq 0$) without composition is not sufficient to upper bound the hockey-stick divergence under composition.

---

> > > ### Comment · Reviewer_G1qb · 2023-11-22
> > >
> > > Thank you for the reply and the discussion.
> > >
> > > I understand that that is the case (substitute relation and subsampling without replacement has a dataset-dependent bound in general), and as said, here would be interesting to find tight bounds for that case. And/or as tightly as possible dominating pairs. E.g. that Gaussian case you mentioned earlier would be interesting. If you look at the RDP analysis in that case, my understanding is that it is looser compared to e.g. the Poisson sampling bounds with add/remove relation.
> > >
> > > I believe that that construction by Doroshenko et al. would give a dominating pair of which hockey-stick divergence would very accurately give the privacy profile given in Zhu et al. (Proposition 30). And that it would not be lossy under compositions either. I think that that DP library for computing the (eps, delta)-bounds is here irrelevant, important is the construction for finding dominating pairs for a given privacy profile.
> > >
> > > My point with RDP was that, if there is a pair of datasets such that the mechanism output for them corresponds to a dominating pair, then there is a post-processing also (Blackwell theorem) such that you get the mechanism outputs for other dataset pairs from this dominating pair and thus it would also dominate the Rényi divergence for all orders. If there is no such worst-case dataset pair, then likely the Rényi divergence will also have different worst-case dataset pairs for different orders, and it would likely be cumbersome to find those worst-case pairs for different orders and to do the RDP accounting. So I think that if one has difficulties with the hockey-stick divergence when trying to determine the dominating pairs of distributions, one would also likely encounter difficulties in RDP accounting.
> > >
> > > As said, I think the paper is dealing with a relevant problem and has interesting observations, but would still need more work.

---

> ### Comment · Reviewer_G1qb · 2023-11-22
>
> Ps. it seems that the worst-case data set in that Laplace example dominates for $\varepsilon \geq 0$ whereas there should be dominance for all $\varepsilon \in \mathbb{R}$ for using the composition result.

---

> > ### Author Response · Authors · 2023-11-23
> >
> > Thank you for your replies and the discussion.
> >
> > Yes, the composition result of Zhu et al. requires that the pair of datasets is worst-case for all $\varepsilon \in \mathbb{R}$.

---

### Official Review · Reviewer_xTEX · 2023-10-31

**Soundness:** 4 excellent
**Presentation:** 3 good
**Contribution:** 3 good
**Rating:** 8
**Confidence:** 3

**Summary:**

This paper clarifies common problems with privacy accounting. In particular, the paper shows that rigorous privacy accounting is affected by the method of sampling batches (Poisson subsampling or sampling without replacement). The paper also shows that self-composing worse-case datasets for the uncomposed mechanism is not in general valid.

**Strengths:**

The paper provides some novel theoretical results that develop our understanding of composition and subsampling with DP. I believe with small presentational edits (see weaknesses), this paper could serve as an important reference work for future research on subsampling with DP, as well as DP practitioners.

In general, the presentation is very clear and the authors on the whole provide welcome intuition to support the mathematical results.

**Weaknesses:**

I am concerned that the paper presents somewhat of a straw-man (e.g. the two points of 'common' confusion in the abstract). It would be useful to provide evidence (even something anecdotal
) that these are common pitfalls in practice. This would make the overall contribution of the paper much more convincing.

The two recommendations for practitioners in the discussion section are welcome but I believe spelling out the implications of this work for practitioners (who will not read through the theory in detail) merits an entire section of the paper, and would enhance its practical utility and potential impact.

Definitions 3-5 would each benefit from a one sentence explanation for those less familiar with the prior research.

**Questions:**

It would be interesting to understand whether empirical privacy (e.g. measured via auditing/MIA) of Poisson subsampling and sampling with replacement differs as much as the theoretical analysis implies (e.g. epsilon =1 vs 10!). Do you have priors about this?

---

> ### Author Response · Authors · 2023-11-18
>
> Thank you for your review.
>
> > I am concerned that the paper presents somewhat of a straw-man (e.g. the two points of 'common' confusion in the abstract). It would be useful to provide evidence (even something anecdotal ) that these are common pitfalls in practice. This would make the overall contribution of the paper much more convincing.
>
> When writing the paper, we made a deliberate decision to minimize drawing attention to existing errors, though we understand how this treatment can risk presenting a straw-man. There are multiple examples of papers that perform privacy accounting for Poisson subsampling when it does not match the implementation. Anecdotally, we can refer the reviewers to page 23 of (De et al., https://arxiv.org/pdf/2204.13650.pdf): “As a minor technical note, we remark that this accountant assumes that at each iteration mini-batches are sampled with replacement from the entire dataset by including every training example with probability 𝑞, while in practice we sample mini-batches using a random shuffling scheme, such that each example is sampled once per training epoch.” Poisson subsampling is computationally inefficient for large datasets and therefore the implementation might instead perform subsampling without replacement or shuffling. We do not want to call out otherwise good works like this in the main body of our paper. We focus on cautioning that such technicalities can in fact lead to large accounting errors. Regarding the pitfall about privacy accountants please see our response to Reviewer G1qb for more details and examples.
>
> > The two recommendations for practitioners in the discussion section are welcome but I believe spelling out the implications of this work for practitioners (who will not read through the theory in detail) merits an entire section of the paper, and would enhance its practical utility and potential impact.
>
> We will expand on implications for practitioners as suggested.
>
> > It would be interesting to understand whether empirical privacy (e.g. measured via auditing/MIA) of Poisson subsampling and sampling with replacement differs as much as the theoretical analysis implies (e.g. epsilon =1 vs 10!). Do you have priors about this?
>
> Please see our response to Reviewer LTUo.

---

### Official Review · Reviewer_LTUo · 2023-10-31

**Soundness:** 3 good
**Presentation:** 3 good
**Contribution:** 3 good
**Rating:** 6
**Confidence:** 3

**Summary:**

This work studies privacy accounting for compositions of subsampled DP mechanisms. The authors show that a pair of worst-case datasets for a subsampled mechanism may no longer be a worst-case after composition, and privacy accounting can be very different for different sub-sampling strategies. These findings call for more care when applying privacy accounting in practice.

**Strengths:**

1. Privacy accounting of DP is an important problem when applying differential privacy in practice.
2. The findings have important practical implications and help to avoid unintended privacy leakage.
3. The theoretical construction of the bad cases is solid.
4. The presentation is clear and easy to follow.

**Weaknesses:**

1. The authors did not provide a viable technical solution for dealing with the worst-case dataset problem under replacement DP.
2. While constructing the bad cases on lower dimensional datasets is sufficient to demonstrate the claims, it would be nice to include empirical results on more realistic datasets to show that these pitfalls can actually appear in practice.

**Questions:**

For replacement DP, is there a good way to find a good approximation of privacy curves when the exact worst-case dataset is hard to obtain?

---

> ### Author Response · Authors · 2023-11-18
>
> We thank the reviewers for their feedback.
>
> > While constructing the bad cases on lower dimensional datasets is sufficient to demonstrate the claims, it would be nice to include empirical results on more realistic datasets to show that these pitfalls can actually appear in practice.
>
> We agree that understanding the privacy loss for actual datasets is an important problem. It would be interesting to e.g. examine how privacy auditing techniques for DP-SGD are affected by various sampling schemes using standard benchmark datasets. It is unlikely that the gradients resemble the worst-case datasets for sampling without replacement in all iterations. That is, all gradients have maximum length and the same direction. It is also unlikely that they resemble the worst-case datasets for Poisson subsampling where all but one datapoint is the zero vector. As such, it is not immediately clear how the sampling schemes compare empirically. However, we consider this a separate problem from tight privacy accounting.
>
> > The authors did not provide a viable technical solution for dealing with the worst-case dataset problem under replacement DP + For replacement DP, is there a good way to find a good approximation of privacy curves when the exact worst-case dataset is hard to obtain?
>
> Determining the worst-case dataset under replacement is a difficult problem in general as we show in Section 7. The technique by Zhu et al. can be used to construct a dominating pair of distributions. See the review and response to Reviewer G1qb for more details. Under subsampling without replacement without composition, the privacy curve of the Gaussian and Laplace mechanisms is the same under the substitution neighborhood as under the remove relation for $\varepsilon \geq 0$. As we show in Sections 5 and 7, that is not the case under composition for the Laplace mechanism. We conjecture that the curve under the substitution and remove relations matches under composition for the Gaussian mechanism. Proving or disproving this is a direction for future work. Nonetheless, the goal of Section 7 is to highlight that accounting for worst-case datasets under substitution is unsolved and that care should be taken when using numerical accounting under the substitution neighboring relation.

---

### Official Review · Reviewer_dGjF · 2023-10-31

**Soundness:** 3 good
**Presentation:** 2 fair
**Contribution:** 3 good
**Rating:** 3
**Confidence:** 3

**Summary:**

This paper exposes several cases where mismatches between privacy accounting and its implementation gives incorrect results. The authors show that the noise required to achieve a certain privacy guarantee can differ significantly between Poisson sampling and sampling without replacement; and that the worst-case dataset for a single iteration of a subsampled mechanism might give incorrect results for the composed mechanism. They also demonstrate issues with computing tight DP bounds under the substitution relation of neighboring datasets.

**Strengths:**

The paper’s findings could be highly impactful on the practice of privacy accounting, as they demonstrate that a method’s implementation should match its accounting. The paper also includes a good call-to-action on how these issues can be addressed by DP practitioners. And despite this paper being about pointing out mistakes in other works, the authors of the paper are tactful and include some interesting discussions in the paper.

**Weaknesses:**

While I think this paper has a powerful message, there are issues with presentation / rigor which make me question whether it’s ready for publication. For example, Proposition 11 is proved by a picture (Figure 1). I also feel that it’s hard to follow the discussion in Section 7 because the counterexample is for something that is stated without any great formality.

The paper is also a bit sparse (there is no appendix). I would have liked to see more substance. The paper does in my opinion do something very important, but in its current state I don't feel it has the solidity of an ICLR paper.

**Questions:**

In Theorem 10, I think there is a bit of a typo: $(1 - \gamma)P + Q$ should be $(1 - \gamma)P + \gamma Q$.

The proof of Proposition 9 seems to take for granted that we know the dominating pair for the Gaussian mechanism! But I think this should be stated formally somewhere as otherwise the “Now, from Theorem 10 we know that…” sentence is unclear.

Combining the plot and the table into a single Figure 3 seems a bit ambitious. At first glance I thought that they were related somehow, and only after reading the caption realized that they were not. I think it would be clearer to separate them into different figures.

The plot in Figure 3 could maybe also be split into two plots? It is a little hard to see the takeaways as is. It might be nice to have one plot showing Poisson for $\epsilon \in [1, 2, 5, 10]$ (to illustrate the “two regions” of high / low sampling rate) and another plot showing Poisson + WOR for $\epsilon=10$ (to illustrate the “hinge”).

---

> ### Author Response · Authors · 2023-11-18
>
> Thank you for the helpful feedback on writing and presentation. We will make several minor edits to address typos and unclear exposition, and we will improve the rigor based on your suggestions. For Proposition 9 we will include a reference for the dominating pair of the Gaussian mechanism (Balle and Wang, https://arxiv.org/pdf/1805.06530.pdf). For Proposition 11 we will include exact calculations for two points on the center plot which is sufficient for the proof of the counter-example. We will rephrase parts of Section 7 and redesign Figure 3 to improve presentation.

---

### Meta-Review · Area_Chair_kmao · 2023-12-12

**Metareview:**

This work addresses two mistakes and subtle points about DP accounting.
1) For composition of subsampled mechanism no single pair of datasets can be used to compute the entire privacy curve
This has been incorrectly assumed in some existing implementations but can be fixed using existing techniques of a dominating pair of distributions
2) A number of papers use accounting for Poisson subsampling whereas (for efficiency reasons) subsampling without replacement (WOR) is used in the actual implementation. This work shows that accounting bounds can be much worse for WOR subsampling
In addition the work shows that for the replacement notion of neighborhood there is in general no dominating pair of datasets.
These are valuable points that merit publication in some form. At the same time there is little new research in this work beyond highlighting the numerics of a known discrepancy (2) and pointing out a mistake in an existing implementation (1). I believe that neither of those contributions is quite at the level of a top ML conference.

**Justification For Why Not Higher Score:**

see above

**Justification For Why Not Lower Score:**

N/A

---

### Decision · Program_Chairs · 2024-01-16

Reject